# Robust Operating Room Scheduling Model with Violation Probability Consideration under Uncertain Surgery Duration

**DOI:** 10.3390/ijerph192013685

**Published:** 2022-10-21

**Authors:** Yanbo Ma, Kaiyue Liu, Zheng Li, Xiang Chen

**Affiliations:** 1School of Management Science and Engineering, Shandong University of Finance and Economics, Jinan 250014, China; 2Department of Nuclear Medicine, Qilu Hospital of Shandong University, Jinan 250012, China

**Keywords:** OR scheduling model, uncertainsurgery duration, constraint violation probability, patients’ waiting time, total OR management cost

## Abstract

This paper proposes an operating room (OR) scheduling model to assign a group of next-day patients to ORs while adhering to OR availability, priorities, and OR overtime constraints. Existing studies usually consider OR scheduling problems by ignoring the influence of uncertainties in surgery durations on the OR assignment. In this paper, we address this issue by formulating accurate patient waiting times as the cumulative sum of uncertain surgery durations from the robust discrete approach point of view. Specifically, by considering the patients’ uncertain surgery duration, we formulate the robust OR scheduling model to minimize the sum of the fixed OR opening cost, the patient waiting penalty cost, and the OR overtime cost. Then, we adopt the box uncertainty set to specify the uncertain surgery duration, and a robustness coefficient is introduced to control the robustness of the model. This resulting robust model is essentially intractable in its original form because there are uncertain variables in both the objective function and constraint. To make this model solvable, we then transform it into a Mixed Integer Linear Programming (MILP) model by employing the robust discrete optimization theory and the strong dual theory. Moreover, to evaluate the reliability of the robust OR scheduling model under different robustness coefficients, we theoretically analyze the constraint violation probability associated with overtime constraints. Finally, an in-depth numerical analysis is conducted to verify the proposed model’s effectiveness and to evaluate the robustness coefficient’s impact on the model performance. Our analytical results indicate the following: (1) With the robustness coefficient, we obtain the tradeoff relationship between the total management cost and the constraint violation probability, i.e., a smaller robustness coefficient yields remarkably lower total management cost at the expense of a noticeably higher constraint violation probability and vice versa. (2) The obtained total management cost is sensitive to small robustness coefficient values, but it hardly changes as the robustness coefficient increases to a specific value. (3) The obtained total management cost becomes increasingly sensitive to the perturbation factor with the decrease in constraint violation probability.

## 1. Introduction

Operating rooms (ORs) are among the principal resources of hospital facilities, accounting for up to 40% of total hospital cost and contributing significantly to hospital revenue [1]. Its scheduling processes are generally rather complicated and particularly expensive regarding the involved procedures and hospital resources. Nevertheless, the practical usage of ORs can acquire many benefits for the hospital, such as shortening patients’ waiting time, increasing patients’ admission, enhancing response time, and improving hospitals’ profitability [2]. To this end, the OR scheduling problem recently received much attention due to its potential to exploit performance gains by utilizing OR resources. It is an essential task for the organization and management of an operating theatre [3,4].

However, devising effective-use OR scheduling faces two significant challenges due to the inherent complexity of the OR scheduling problem [5]. The first is formulating the performance metric related to patients’ waiting time and OR overtime. Long waiting times are leading causes of patients’ dissatisfaction, which not only essentially decrease OR costs and surgical outcomes but also harm patients’ psychological status [6]. Hence, an accurate measure of patients’ waiting time should be provided as necessary for an OR scheduling model [7]. Meanwhile, some studies [8] have shown that the opening time of OR greatly exceeds regular working hours, sometimes even to 14 h, which is associated with increased medical errors, dissatisfaction among scarce staff, etc. The second is addressing the uncertainties in surgical procedure duration performed on patients. This surgery duration usually involves a high degree of uncertainty due to factors such as each patient’s physical condition, physicians’ level of expertise, and gaps in the availability of surgical equipment. The neglect of such uncertainty when formulating OR scheduling model will result in prolonged patient waiting times, reducing the utilization of medical resources and harming OR costs and surgical outcomes. In order to address these issues and fully exploit the potential performance gains of utilizing the OR resource, we focus on the OR scheduling problem by considering uncertain surgery durations to optimize patients’ waiting time and OR overtime.

Thus far, much effort has been made concerning the performance metric addressing OR scheduling problems by formulating a combinatorial optimization model. The goal of the scheduling problem is to allocate a given amount of resources to achieve a certain performance metric, such as maximizing the utilization rates of the OR, minimizing the operational cost, minimizing OR overtime, minimizing patients’ waiting time, etc. [6,7,9,10,11,12,13,14,15,16,17,18,19,20,21,22,23,24,25,26,27,28,29]. For example, Mancilla et al. [9] developed a stochastic model to minimize the weighted cost associated with patients’ waiting time, OR idle time, and overtime. Moreno and Blanco [10] designed an operating procedure that maximizes OR utilization rates and minimizes overtime cost. Khaniyev et al. [11] studied the OR scheduling problem with the optimization objective of minimizing the weighted sum of expected patients’ waiting time, room availability, and overtime. It is worth noting that minimizing patients’ waiting time is also considered an optimization objective [7] aiming to shorten patients’ waiting time. Among the available studies on minimizing patients’ waiting time, Addis et al. [12] minimized the penalty cost generated by patients waiting for surgeries in a week; in this week, patients may have surgery or not. Erdogan and Denton [13] addressed the problem of determining the optimal time for a patient to undergo surgery to minimize the weighted sum of expected patients’ waiting time and determine a definitive appointment schedule in a cycle. Issabakhsh et al. [14] considered the scheduling problem to minimize total patients’ waiting time and the weighted sum of all patients’ completion time in a cycle. Castaing et al. [15] presented a new two-stage stochastic integer model capable of optimizing OR overtime and expected patients’ waiting time in a week. However, these schemes calculated patients’ waiting time in days rather than using accurate measures of patients’ surgery duration, thereby neglecting the influence of the duration of every surgery on that day in the OR scheduling model. Moreover, this neglect of the patients’ surgery duration in calculating the patients’ waiting time will heighten the gap between theoretical research and management practices. In this paper, by conceiving the patients’ waiting sequence for each OR, we specify the accurate patients’ waiting time by summing the surgery’s duration and formulating the patients’ waiting costs to investigate the quantitative relationship between patients’ waiting time and OR scheduling model.

Apart from performance metrics, many studies have been conducted on designing OR solutions that address uncertain surgery duration. Thereinto, three approaches are prevalent for addressing the uncertain surgery duration in discrete variable models, including stochastic programming [15,16,17], fuzzy optimization [18,19], and robust optimization [20,21,22,23,24,25,26,27]. For example, Castaing et al. [15] proposed a two-stage stochastic programming model to design a new appointment schedule for patients with uncertain surgery duration. Lamiri et al. [16] proposed a stochastic programming model for OR scheduling systems in the presence of elective and emergency surgeries. Lahijanian et al. [18] considered each patient’s surgery duration as a fuzzy number and investigated the OR scheduling problem using a fuzzy optimization method. Wang et al. [19] used fuzzy numbers to represent patients’ uncertain surgery duration and applied a new hybrid meta-heuristic and fuzzy optimization to obtain detailed OR scheduling. Although both stochastic programming and fuzzy optimization can process uncertain parameters in discrete variable models, the optimality of their proposed solution cannot be preserved when the data change. In particular, stochastic programming requires the knowledge that uncertain parameters obey a particular probability distribution, which is difficult to obtain in reality. In recent years, robust optimization has been widely used as an effective tool to address uncertain decision problems. This method does not constrain the specific distribution form of uncertain parameters, requiring only the mean and variance from historical data. At the same time, the resulting solution can remain feasible in a changing environment. The robust optimization theory was proposed by Soyster et al. [30] and then in-depth studies were conducted by Ben-Tal, Bertsimas, and others [31,32,33,34], which established a solid foundation for robust optimization theories. Therein, a robust discrete approach proposed by Bertsimas in 2004 [34] has been widely used to formulate OR scheduling models [20,21,22,23,24,25,26,27]. By introducing the robustness coefficient, this approach allows the level of robustness to be controlled in an OR scheduling model, i.e., the effect of the worst-case approach is moderated [35]. For example, Denton et al. [23] applied robust optimization to solve the OR scheduling problem in 2010, establishing a robust optimization model under patients’ uncertain surgery durations. Breuer et al. [20] integrated the needs of multiple stakeholders to develop a robust model combining staffing and scheduling decisions in OR. Amirhossein et al. [21] used a local neighbourhood search algorithm to find the optimal solution of a model in a multi-objective robust decision model considering upstream and downstream tasks. Wang et al. [22] considered each patient’s surgery duration as a bounded interval and developed a two-phase robust optimization method. Peng et al. [27] proposed a two-stage robust surgical planning and scheduling model using an ellipsoidal uncertainty set and box uncertainty set to portray uncertain surgical duration and the postoperative ICU length of stay, respectively. The above research studies provide a trade-off between robustness and optimal solutions via regulating the robustness coefficient, which distinctly increases the system’s performance compared with the worst-case solution. Evidently, this increase in system performance comes at the cost of increasing the violation probability of the resource constraint. However, none of the above research quantitatively analyzes the relationship between the constraint violation probability and system performance. In this paper, we will develop a robust OR scheduling model with a robustness coefficient to manage uncertainties in the surgical duration and to investigate the constraint violation probabilities under different robustness coefficients by invoking the robust discrete approach.

Against this background, the present study aims to develop and evaluate a robust scheduling model to determine opening OR and overtime and assigns a group of next-day patients to ORs while adhering to OR availability, priorities, and overtime constraints. By considering the patients’ uncertain surgery duration, we formulate a robust OR scheduling model designed to minimize the sum of the fixed OR opening cost, the patients’ waiting penalty cost, and the OR overtime cost. The proposed model is based on robust discrete optimization for studying the influence of uncertain surgery durations on the OR scheduling system to support "next-day hospital"-type decision making. The contributions of this paper are summarized as follows:▲The first contribution of this paper relates to the novelty of the optimization objective, which is formulated as the sum of the fixed OR opening cost, the patients’ waiting penalty cost, and the OR overtime cost to represent the operational cost of ORs. Thereinto, patients’ waiting times are calculated via the cumulative sum of the accurate patient’s surgery duration. Hence, with this formula, we demonstrate the influence of the uncertain surgery duration on the OR scheduling model.▲Based on the developed optimization objective, we formulate the robust OR scheduling model by considering the patient’s uncertain surgery duration. Thereinto, the uncertain surgery duration is represented by a box uncertainty set, and a robustness coefficient is introduced to control the trade-off between the constraint violation probability and optimality. Then, the robust discrete optimization theory and strong dual theory are invoked to transform the robust model equivalently into a Mixed Integer Linear Programming (MILP) model, which is in a tractable analytical form. It is noteworthy that this paper solves the robust optimization problem with uncertain parameters in both the objective function and constraints. Moreover, by invoking the robust discrete theory, we derive the probability bounds of constraint violation for the OR overtime constraints, which is one of the early attempts to study the constraint violation probability based on the theoretical analysis of the OR scheduling model.▲Third, we validate the performance of our model by calculating the upper bound of the constraint violation probability and the objective function at different robustness coefficients to investigate the trade-off among robustness, optimality, and reliability. In addition, some insights are provided on the influence of the robustness coefficient on the OR scheduling model. A sensitivity analysis of the uncertain perturbation factor is also performed to obtain variations in hospital cost sensitivity with the constraint violation probability.

The remainder of this study is organized as follows. We describe the problem and state the assumptions involved in the issue in Section 2. In Section 3, we build a deterministic model and propose a robust model for next-day surgery. Then, we convert it into an MILP model and suggest an upper probability bound on the likelihood of constraint violations under various robustness coefficients. We describe the results of numerical calculations and provide a sensitivity analysis of perturbation factors in Section 4. We conclude the paper and provide some recommendations for hospital managers in Section 5.

## 2. OR Model and Assumptions

In this study, we consider a next-day OR scheduling model. With a given set of patients and a homogeneous set of ORs, this scheduling model helps assign patients to ORs and makes decisions regarding OR working times and overtime.

In this OR scheduling model, we suppose *n* ORs are available for *T* regular opening hours each day, and they provide homogeneous services for all patients on the waiting sequence. We assume that the orthogonal OR sharing mode is performed; i.e., each OR can only be assigned to one patient at a given time, and each patient is operated on in only one OR. $x_{ij}\in \left\{0,1\right\}$ is an indicator factor for OR allocation; i.e., $x_{ij}=1$ means that the *i*-th patient is assigned in the *j*-th OR for surgery; otherwise, $x_{ij}=0$. Any procedure performed in an OR incurs different costs, including the fixed opening cost incurred during regular opening hours and additional costs incurred during overtime opening hours. In this paper, we argue that the OR can extend beyond regular opening hours but note that OR overtime costs are much higher than fixed opening costs as physicians and nurses become less productive during overtime. We denote the unit OR overtime cost as $r_{j}$ and the overtime of the *j*-th OR as $o_{j}$, where $o_{j}\geq 0$ and $o_{j}=0$ when there are no overtime hours.

We assume that there are *m* patients waiting for procedures, and the surgery duration of the *i*-th patient is denoted as $d_{i}$. Every patient’s surgery duration varies independently from one patient to another due to their medical conditions or physical status differences. The patients’ sequence is generated based on the order of the initial patients’ appointments, and we use M={1,2,…,i,…,m} to denote the index set of patient. Furthermore, all patients are presumed to be time-sensitive because long patient waiting times are detrimental to their condition and psychological state [36]. Thus, the patient’s waiting cost is incurred for the hospital, which is related to patients’ waiting time and patients’ time sensitivity. Here, we introduce $b_{i}$ to denote the weight of the *i*-th patient’s waiting time, which is set in the interval $\left[1,3\right]$ by a physician or nurse practitioner according to the state of the illness. We can obtain the patients’ waiting penalty cost by multiplying the per-patient waiting time weight and each patient’s specific waiting time. Mathematically, it can be denoted as $\sum_{i=1}^{m}b_{i}x_{ij}\left(\sum_{k=1}^{i-1}d_{k}x_{kj}\right)$. For the ease of representation, the key parameters in this paper are summarized in Table 1.

For the hospital, the objective of resource allocation is to make decisions about the opening OR and ORs overtime and to assign OR to each patient. Hence, this minimizes the total OR management cost, defined as the sum of fixed OR opening cost, OR overtime cost, and patients’ waiting penalty cost. Mathematically, we formulate this objective as follows:$$\begin{array}{c} \sum_{j=1}^{n}\left\{c_{j}y_{j}+\sum_{i=1}^{m}b_{i}x_{ij}\left(\sum_{k=1}^{i-1}d_{k}x_{kj}\right)+r_{j}y_{j}o_{j}\right\},\forall i\in M,j\in N, \end{array}$$
where the patients’ waiting penalty cost is incurred by patients’ waiting time, $y_{j}$ is a 0−1 variable, and $y_{j}=1$ if *j*-th OR is open; otherwise, $y_{j}=0$.

## 3. Robust OR Scheduling Approach

In this section, we first construct a deterministic OR scheduling model based on the assumption that all patients’ accurate surgery durations are available. Then, we present the robust counterpart formulation of the OR scheduling model and show how we transform it into an MILP problem by employing the robust discrete optimization theory and the strong dual theory. Afterwards, we provide the upper-bound formula for the violation probability of overtime constraints.

Naturally, we recapitulated the research ideas of Section 3 to a flowchart as shown in Figure 1.

### 3.1. Deterministic Model

We present the OR scheduling model with a set of homogeneous ORs for all patients in the waiting sequence by developing a formulation for a deterministic model. Assuming that the patient’s surgery duration is known precisely for each patient, this optimization problem can be mathematically formulated to minimize the total OR management cost while satisfying medical resource constraints as follows: (1)$$\begin{array}{ccc} P1: & \min & \sum_{j=1}^{n}\left[c_{j}y_{j}+\sum_{i=1}^{m}b_{i}x_{ij}\left(\sum_{k=1}^{i-1}d_{k}x_{kj}\right)+r_{j}y_{j}o_{j}\right], \end{array}$$(2)$$\begin{array}{ccc} & s.t. & \sum_{j=1}^{n}x_{ij}=1,\forall i\in M, \end{array}$$(3)$$\begin{array}{ccc} & & x_{ij}\leq y_{j},\forall i\in M,j\in N, \end{array}$$(4)$$\begin{array}{ccc} & & \sum_{i=1}^{m}d_{i}x_{ij}\leq Ty_{j}+o_{j},\forall j\in N, \end{array}$$(5)$$\begin{array}{ccc} & & x_{ij}\in \left\{0,1\right\},\forall i\in M,j\in N, \end{array}$$(6)$$\begin{array}{ccc} & & y_{j}\in \left\{0,1\right\},\forall j\in N, \end{array}$$(7)$$\begin{array}{ccc} & & o_{j}\geq 0,\forall j\in N. \end{array}$$

The OR scheduling optimization problem P1 is a deterministic model since the surgery duration is assumed to be estimated precisely for each patient. Therein, the objective function in Equation (Equation 1) aims to minimize the total OR management cost. The constraint in Equation (Equation 2) indicates that each patient must be allocated to an OR to guarantee that all surgeries must be completed. The constraint in Equation (Equation 3) suggests that surgical procedures can only be performed in open ORs for patients. The constraint in Equation (Equation 4) constrains the sum of fixed working time and overtime for the *j*-th OR under a specified threshold specified by $T+o_{j}$.

The setup described in the objective function in Equation (Equation 1) requires knowledge of the surgery duration of each patient, i.e., $d_{i}$ in all i∈M is assumed to be known by the decision maker. However, in practice, there exists uncertainty regarding the surgery’s duration due to the inherent variability of surgery. Thus, the optimal solution for the OR scheduling model with such a setup is not valid; it would lead to long patient waiting times and an unreasonable allocation of medical resources. In the following subsection, we consider this uncertainty by using the box uncertainty set and developing a robust OR scheduling model to accommodate the uncertainty.

### 3.2. Robust Discrete Optimization Model and Transformation

In this subsection, we use the robust discrete optimization theory to construct a robust model by considering the uncertain surgery duration. Firstly, we use the box uncertainty set to represent the fluctuation range of the value of random variable $d_{i}$, which can help us describe the uncertain surgery duration that appears in the objective function (Equation 1) and OR overtime constraints (Equation 4). Assume that $d_{i}$ is the actual surgery duration for patient *i*. The bounded box uncertainty set for the random variable $d_{i}$ is denoted as $\left[\bar{d_{i}}-\hat{d_{i}},\bar{d_{i}}+\hat{d_{i}}\right]$, where $\bar{d_{i}}$ denotes the mean value of the uncertain surgery duration and $\hat{d_{i}}$ denotes the maximum deviation from $\bar{d_{i}}$. By the way, mean value $\bar{d_{i}}$ and maximum deviation $\hat{d_{i}}$ are provided by previous empirical data or the treating physician. According to the robust discrete optimization theory [34], proportional deviation $\frac{|d_{i}-\bar{d_{i}}|}{\hat{d_{i}}}$ is introduced, which can be interpreted as the deviation of $d_{i}$ from the mean, $\bar{d_{i}}$. Moreover, we assume that the proportional deviation $\frac{|d_{i}-\bar{d_{i}}|}{\hat{d_{i}}}$ for different patients undergoes independent distributions in the range [0, 1]. In particular, when $\frac{|d_{i}-\bar{d_{i}}|}{\hat{d_{i}}}=0$, $d_{i}=\bar{d_{i}}$; when $\frac{|d_{i}-\bar{d_{i}}|}{\hat{d_{i}}}=1$, $d_{i}=\bar{d_{i}}\pm \hat{d_{i}}$.

To flexibly regulate the conservatism of the worst-case solution [37], robustness coefficient Γ is introduced to control the level of robustness representing the trade-off between optimality and robustness. Specifically, we assume that Γ∈[0,m] and Γ can be taken as a fractional number. Γ=0 corresponds to the case in which the accurate surgery duration of each patient is unrealistically assumed; i.e., the uncertain surgery duration $d_{i}$ is taken as the mean value. For Γ=m, we consider the uncertain surgery duration for *m* patients where $d_{i},\forall i$, takes the endpoints of the interval $\left[\bar{d_{i}}-\hat{d_{i}},\bar{d_{i}}+\hat{d_{i}}\right]$, which is actually the worst-case scenario for the OR scheduling model. From the above description of Γ, we can infer that adjusting the value of Γ can control the conservatism of the worst-case solution. Specifically, the larger the value of Γ, which means more uncertainty parameters, the stronger the robustness. Then, the resulting solution is more conservative. However, such conservatism will sacrifice part of the system’s performance, i.e., increasing the total management cost and vice versa. Hence, adjusting parameter Γ is feasible for achieving the trade-off between robustness and conservatism. In summary, the OR scheduling model with such a coefficient Γ can provide a flexible solution to handle uncertain parameters. In particular, when assigning *m* to Γ, the OR scheduling model with Γ reduces to the worst-case model for the OR scheduling model.

Based on the definitions of the random variable $d_{i}$ and robustness coefficient Γ, we can obtain the following.
$$\sum_{i=1}^{m}\frac{|d_{i}-\bar{d_{i}}|}{\hat{d_{i}}}\leq \Gamma .$$

Thus, the uncertainty set U(Γ) for the uncertain surgery duration $d_{i}$ can be expressed as follows: (8)$$U\left(\Gamma \right)=\left\{d|d_{i}\in \left[\bar{d_{i}}-\hat{d_{i}},\bar{d_{i}}+\hat{d_{i}}\right],\forall i\in M;\sum_{i=1}^{m}\frac{|d_{i}-\bar{d_{i}}|}{\hat{d_{i}}}\leq \Gamma \right\}.$$

d and x represent vectors or matrices in the following. From Equation (Equation 8), we can observe that uncertainty set U(Γ) is associated not only with the random variable $d_{i}$ but also with robustness coefficient Γ, which is in charge of adjusting the level of robustness. With the uncertainty set of d being U(Γ), we aim to design an OR scheduling model for minimizing the total management cost, which can be written as follows: (9)$$\begin{array}{ccc} P2: & \min & \sum_{j=1}^{n}\left\{c_{j}y_{j}+\psi \left(x\right)+r_{j}y_{j}o_{j}\right\},\forall i\in M,\forall j\in N,\\ & s.t. & (2),\;(3),\;(5),\;(6),\;(7), \end{array}$$(10)$$\begin{array}{ccc} & & \sum_{i=1}^{m}d_{i}x_{ij}\leq Ty_{j}+o_{j},\forall j\in N, \end{array}$$(11)$$\begin{array}{ccc} & & d_{i}\in U\left(\Gamma \right),\forall i\in M, \end{array}$$
where ψ(x) is the worst-case waiting penalty cost among all surgery duration possibilities, which is defined by the following:$$\begin{array}{c} \psi \left(x\right)=\max_{d\in U(\Gamma )}\sum_{i=1}^{m}b_{i}x_{ij}\left(\sum_{k=1}^{i-1}d_{k}x_{kj}\right). \end{array}$$

The robust model P2 is a conservative one from which we can observe that the total OR management cost can be guaranteed to be no less than the worst-case optimal solution to Equation (Equation 9) for all surgery possibilities under the given U(Γ). The significance of the robust model is that the inclusion of Γ allows us to find a solution that can adjust the degree of robustness, showing a significant advantage in terms of various risks while balancing robustness and optimality. It is worth noting that problem P2 is the nominal one of deterministic model formulation, where perfect surgery duration information is assumed to be available; i.e., the estimated values are considered as exact values.

In general, solving the robust optimization problem involves high computational complexity. This will further aggravate our proposed model, where there exists uncertain surgery durations in both the objective function and overtime constraints. To reduce the computational complexity of solving P2, we treat $d_{i}$ as a bounded random variable and transform the objective function based on $\bar{d_{i}}$ and $\hat{d_{i}}$. Moreover, we will also demonstrate that the proposed reformulation provides a trade-off between optimality and robustness by using Γ. Then, to further simplify formulations, we will utilize the notion of a protection function to transform overtime constraints. Based on these transformations, a tractable formulation for the proposed OR scheduling model is obtained such that P2 can be solved with no additional computational complexity.

#### 3.2.1. Linearization of the Objective Function

Evidently, the direct way to solve optimization problem P2 is to obtain maximization function ψ(x) under uncertainty set (Equation 8) analytically and then solve the outer minimization in P2. In this manner, we first consider the maximization function under uncertainty set (Equation 8), in which the number of random variables di is Γ. We denote Ji=1,…,i−1,i∈N as the index set of patients in front of the *i*-th patient, i.e., for each *i*
Ji consists of patients from 1 to i−1 in the waiting sequence. To obtain a tractable analytical form of φ(x), we divide uncertain parameters into two parts, i.e., the integer part Γ, which is the value after rounding down Γ, and fractional part Γ−Γ. In addition, denote *S* as a subset of set M, i.e., S⊆M, and subset *S* satisfies S=Γ, where S is the number of elements in subset *S*, and Γ is the value after rounding down Γ. Additionally, *r* is a subscript that satisfies r∈M\S, which represents an element remaining in set M after removing the elements in the set S. Then, this part of the internal maximization problem can be transformed as follows:(12)ψ(x)=maxd∈U(Γ)∑i=1mbixij(∑k=1i−1dkxkj)=maxS∪r∣S⊆M,S=Γ,r∈M/S∑i=1mbixij∑k∈S∩Jid^kxkj+∑i=r+1mbixijd^rΓ−Γxrj+∑i=1mbixij(∑k=1i−1d¯kxkj).

From Equation (Equation 12), we can see that ψ(x) comprises two parts: the maximization function part, which is related with ${\hat{d}}_{i}$, and the deterministic part, which is not related to uncertainties in surgery durations. In particular, when ⌊Γ⌋ is an integer, the deterministic part would not exist, and then the maximization problem is reduced to the following:$$\begin{array}{cc} \; & \max_{d\in U(\Gamma )}\sum_{i=1}^{m}b_{i}x_{ij}\left(\sum_{k=1}^{i-1}d_{k}x_{kj}\right)=\\ \; & \max_{\left\{S\cup \left\{r\right\}|S\subseteq M,\left|S\right|=\left\lfloor \Gamma \right\rfloor ,r\in M/S\right\}}\left\{\sum_{i=1}^{m}b_{i}x_{ij}\sum_{k\in S\cap J_{i}}{\hat{d}}_{k}x_{kj}\right\}+\sum_{i=1}^{m}b_{i}x_{ij}\left(\sum_{k=1}^{i-1}{\bar{d}}_{k}x_{kj}\right). \end{array}$$

Then, we derive the following theorem to characterize how the maximization problem in Equation (Equation 12) is transformed into a linear optimization problem.

**Theorem** **1.**
*We introduce variables $w_{i}$, and the following is a maximization problem (Equation (Equation 12)):*

$$\max_{\left\{S\cup \left\{r\right\}|S\subseteq M,\left|S\right|=\left\lfloor \Gamma \right\rfloor ,r\in M/S\right\}}\left\{\sum_{i=1}^{m}b_{i}x_{ij}\sum_{k\in S\cap J_{i}}{\hat{d}}_{k}x_{kj}+\sum_{i=r+1}^{m}b_{i}x_{ij}\left({\hat{d}}_{r}\left(\Gamma -\left\lfloor \Gamma \right\rfloor \right)x_{rj}\right)\right\}$$

*which is equivalent to the following linear optimization problem.*

(13)
$$\begin{array}{ccc} & \max & \sum_{i=1}^{m}b_{i}x_{ij}\left(\sum_{k=1}^{i-1}{\hat{d}}_{i}x_{kj}w_{i}\right), \end{array}$$


(14)
$$\begin{array}{ccc} & s.t. & \sum_{i=1}^{m}w_{i}\leq \Gamma , \end{array}$$


(15)
$$\begin{array}{ccc} & & 0\leq w_{i}\leq 1,\forall i\in M. \end{array}$$



**Proof.** According to Bertsimas [34], the optimal $\omega_{i}$ of Equations (Equation 13)–(Equation 15) comprises ⌊Γ⌋ variables at 1, one variable at (Γ−⌊Γ⌋), and the remainder is 0. Evidently, this optimal solution is equivalent to selecting a suitable subset:
$$\left\{S\cup \left\{r\right\}|S\subseteq M,\left|S\right|=\left\lfloor \Gamma \right\rfloor ,r\in M/S\right\}$$
from set M such that the following is maximized.
$$\left\{\sum_{i=1}^{m}b_{i}x_{ij}\sum_{k\in S\cap J_{i}}{\hat{d}}_{k}x_{kj}+\sum_{i=r+1}^{m}b_{i}x_{ij}\left({\hat{d}}_{r}\left(\Gamma -\left\lfloor \Gamma \right\rfloor \right)x_{rj}\right)\right\}$$This completes the proof. □

By using the theorem mentioned earlier, we can transform the optimization problem as follows.
(16)$$\begin{array}{ccc} & \min & \sum_{j=1}^{n}\left\{c_{j}y_{j}+\sum_{i=1}^{m}b_{i}x_{ij}\sum_{k=1}^{i-1}{\bar{d}}_{i}x_{kj}+\max\sum_{i=1}^{m}b_{i}x_{ij}\sum_{k=1}^{i-1}{\hat{d}}_{k}x_{kj}w_{i}+r_{j}y_{j}o_{j}\right\},\\ & s.t. & (2),\;(3),\;(5),\;(6),\;(7),\;(10),\;(14),\;(15). \end{array}$$

However, the resulting model (Equation 16) is still difficult to solve directly due to the nonlinearity of the objective function in (Equation 16). Considering that the optimization problem comprising Equations (Equation 13)–(Equation 15) is bounded and feasible, we transform the max-function in Equation (Equation 16) into the min-function by employing the dual theory. With this transformation, the linearization of the objective function can be achieved. Note that according to the duality theory [34], the strong duality the primal problem in (Equation 13)–(Equation 15) holds. Thus, the primal problem can be equivalently solved by solving the dual problem. Mathematically, the dual problem can be expressed as follows: (17)$$\begin{array}{ccc} & \min & \Gamma h+\sum_{i=1}^{m}g_{i}, \end{array}$$(18)$$\begin{array}{ccc} & s.t. & h+g_{i}\geq b_{i}x_{ij}\sum_{k=1}^{i-1}{\hat{d}}_{k}x_{kj},\forall i\in M,\forall j\in N, \end{array}$$(19)$$\begin{array}{ccc} & & h,g_{i}\geq 0,\forall i\in M, \end{array}$$
where *h* denotes the dual variable associated with constraint $\sum_{i=1}^{m}w_{i}\leq \Gamma$, and dual variable $g_{i}$ corresponds to constraints $0\leq w_{i}\leq 1$. With this dual problem, the linearized form of the model (Equation 16) ends up as follows:(20)$$\begin{array}{ccc} & \min & \sum_{j=1}^{n}\left\{c_{j}y_{j}+\sum_{i=1}^{m}b_{i}x_{ij}\sum_{k=1}^{i-1}{\bar{d}}_{i}x_{kj}+\Gamma h+\sum_{i=1}^{m}g_{i}+r_{j}y_{j}o_{j}\right\},\\ & s.t. & (2),\;(3),\;(5),\;(6),\;(7),\;(10),\;(18),\;(19). \end{array}$$

Thus far, the linearization of objective function (Equation 16) is compelted.

#### 3.2.2. The Overtime Constraints Transformation

It is time to consider the uncertain surgery duration in the overtime constraints in Equation (Equation 10). The overtime constraints in Equation (Equation 10) are satisfied if and only if the following is the case:$$\begin{array}{c} \max_{d\in U(\Gamma )}\sum_{i=1}^{m}d_{i}x_{ij}\leq Ty_{j}+o_{j},\forall j\in N, \end{array}$$
which is equivalent to the following:(21)$$\begin{array}{c} \sum_{i=1}^{m}{\bar{d}}_{i}x_{ij}+\max_{d\in U(\Gamma )}\sum_{i=1}^{m}\left(d_{i}-{\bar{d}}_{i}\right)x_{ij}\leq Ty_{j}+o_{j},\forall j\in N. \end{array}$$

With the definition of U(Γ), we can derive that constraint Equation (Equation 21) can be equivalently transformed by the following constraint for each OR *j*.
(22)$$\beta_{j}\left(x,\Gamma \right)+\sum_{i=1}^{m}{\bar{d}}_{i}x_{ij}\leq Ty_{j}+o_{j},\forall j\in N.$$

Given a vector x, the protection function
$$\beta_{j}\left(x,\Gamma \right)=\max_{\left\{S\cup \left\{r\right\}|S\subseteq M,\left|S\right|=\left\lfloor \Gamma \right\rfloor ,r\in M/S\right\}}\left\{\sum_{i\in S}{\hat{d}}_{i}x_{ij}+\left(\Gamma -\left\lfloor \Gamma \right\rfloor \right){\hat{d}}_{r}x_{ij}\right\}$$
of *j*-th constraint is protected against the uncertainty in surgery duration. As a special case, when Γ takes an integer value, the protection function reduces to the following:$$\beta_{j}\left(x,\Gamma \right)=\max_{\left\{S\cup \left\{r\right\}|S\subseteq M,\left|S\right|=\left\lfloor \Gamma \right\rfloor ,r\in M/S\right\}}\sum_{i\in S}{\hat{d}}_{i}x_{ij}.$$

Notably, similarly to Equation (Equation 12), we can observe that the constraint function in Equation (Equation 22) is constitutive of two parts: the protection function part, which is related to ${\hat{d}}_{i}$, and the deterministic part, which is not related to the uncertainty in surgery durations.

By introducing variables $a_{i},\forall i\in M$, we propose the following Theorem 2 to reformulate constraint Equation (Equation 22) as a linear form.

**Theorem** **2.***The protection function of the *j*-th constraint, $\beta_{j}\left(x,\Gamma \right)$, is equivalent to the following linear optimization problem, which can be written as follows:*(23)$$\begin{array}{ccc} \beta_{j}\left(x,\Gamma \right)= & \max & \sum_{i=1}^{m}{\hat{d}}_{i}x_{ij}a_{i}, \end{array}$$(24)$$\begin{array}{ccc} & s.t. & \sum_{i=1}^{m}a_{i}\leq \Gamma , \end{array}$$(25)$$\begin{array}{ccc} & & 0\leq a_{i}\leq 1,\forall i\in M. \end{array}$$

Here, for ∀i∈M, $a_{i}=1$, at which point Γ=m, and constraint Equation (Equation 22) becomes $\sum_{i=1}^{m}\left({\bar{d}}_{i}+{\hat{d}}_{i}\right)x_{ij}\leq Ty_{j}+o_{j}$. In this case, uncertain surgery durations are all valued at the endpoints of interval $\left[\bar{d_{i}}-\hat{d_{i}},\bar{d_{i}}+\hat{d_{i}}\right]$, which becomes the robust method proposed by Soyster [30]. It may be far from the optimal solution, significantly impacting the value of the objective function.

Due to how strong duality holds, by introducing dual variables *p* and $q_{i}$, the dual problem obtained from the model comprising Equations (Equation 23)–(Equation 25) can be expressed as follows: (26)$$\begin{array}{ccc} & \min & p\Gamma +\sum_{i=1}^{m}q_{i}, \end{array}$$(27)$$\begin{array}{ccc} & s.t. & p+q_{i}\geq {\hat{d}}_{i}x_{ij},\forall i\in M,j\in N, \end{array}$$(28)$$\begin{array}{ccc} & & p,q_{i}\geq 0,\forall i\in M, \end{array}$$
where dual variable *p* corresponds to constraint $\sum_{i=1}^{m}a_{i}\leq \Gamma$ and dual variables $q_{i}$ correspond to constraint $0\leq a_{i}\leq 1$. Because this part appears as a constraint in the original model, we convert objective function (Equation 26) to the constrained form. Thus, Equation (Equation 22) can be rewritten as follows.
(29)$$\begin{array}{cc} & \sum_{i=1}^{m}{\bar{d}}_{i}x_{ij}+p\Gamma +\sum_{i=1}^{m}q_{i}\leq Ty_{j}+o_{j},\forall j\in N,\\ & p+q_{i}\geq {\hat{d}}_{i}x_{ij},\forall i\in M,j\in N,\\ & p,q_{i}\geq 0,\forall i\in M. \end{array}$$

Finally, the robust equivalent model of the next-day OR scheduling model can be obtained as follows.
(30)$$\begin{array}{ccc} & \min & \sum_{j=1}^{n}\left\{c_{j}y_{j}+\sum_{i=1}^{m}b_{i}x_{ij}\left(\sum_{k=1}^{i-1}{\bar{d}}_{k}x_{kj}\right)+\Gamma h+\sum_{i=1}^{m}g_{i}+r_{j}y_{j}o_{j}\right\},\\ & s.t. & \sum_{j=1}^{n}x_{ij}=1,\forall i\in M,\\ & & x_{ij}\leq y_{j},\forall i\in M,j\in N,\\ & & \sum_{i=1}^{m}{\bar{d}}_{i}x_{ij}+p\Gamma +\sum_{i=1}^{m}q_{i}\leq Ty_{j}+o_{j},\forall j\in N,\\ & & h+g_{i}\geq b_{i}x_{ij}\sum_{k=1}^{i-1}{\hat{d}}_{k}x_{kj},\forall i\in M,j\in N,\\ & & p+q_{i}\geq {\hat{d}}_{i}x_{ij},\forall i\in M,j\in N,\\ & & x_{ij}\in \left\{0,1\right\},\forall i\in M,j\in N,\\ & & y_{j}\in \left\{0,1\right\},\forall j\in N,\\ & & h,p,g_{i},q_{i},o_{j}\geq 0,\forall i\in M,j\in N. \end{array}$$

After the above transformation work, the uncertain surgery duration does not exist anymore. Moreover, it can be observed that the robust equivalent model (Equation 30) is an MILP model. However, many constraints make a direct solution of the problem difficult, so we use the Python 3.10 programming language to encode the model (Equation 30) and use the Gurobi 9.1.2 solver to solve it.

### 3.3. Upper-Bound Constraint Violation Probability

Although the optimization method employed in Section 3.2 can guarantee that the solution remains feasible in the worst-case scenario, the employed uncertainty set, U(Γ), does not consider all parameters involved in the uncertain surgery duration, which inherently results in the constraint violation in the optimization model. In a particular case, when Γ=m, the proposed solution guarantees that the optimization constraint is satisfied for every realization in uncertainty set U(Γ) by resorting to the worst-case optimization formulation because the uncertain surgery duration of *m* patients is considered. Thus, we can conclude that although the proposed probability can be developed based on the model in (Equation 30), it can only provide a probability guarantee for the overtime constraints; i.e., the proposed solution can provide a trade-off between conservative performance and probability guarantees for overtime constraints. Moreover, the constraint violation probability is an important performance index reflecting the service level of the OR scheduling model. It is also one of the factors considered by the decision maker of the hospital. Hence, the quantitative relationship between the constraint violation probability and Γ must be discussed. To make it easy, we denote the value of constraint violation probability as *P*.

**Theorem** **3.**
*For a given Γ≥0, the upper probability that a robust solution can satisfy the constraint is expressed as follows:*

(31)
$$P=Pr\left(\sum_{i=1}^{m}d_{i}x_{ij}\leq Ty_{j}+o_{j}\right)\leq \left(1-\mu \right)C\left(m,\left\lfloor v\right\rfloor \right)+\sum_{l=\left\lfloor v\right\rfloor +1}^{m}C\left(m,l\right),$$

*where v=(Γ+m)/2, $\mu =v-\left\lfloor v\right\rfloor$ and the following is the case.*

*

$C\left(m,l\right)=\left\{\begin{array}{c} \frac{1}{2^{m}}\;\;\;\;\;\;\;\;\;\;\;,\;\mathrm{if}\;l=0\;or\;l=m\\ \frac{1}{\sqrt{2\pi }}\sqrt{\frac{m}{(m-l)l}}\cdot exp\left\{m\cdot log\left(\frac{m}{2(m-l)}\right)+l\cdot log\left(\frac{m-l}{l}\right)\right\}\;,\;\mathrm{else}\; \end{array}\right..$

*


Theorem 3 shows the probability that the robust solution satisfies constraints, and by further exploring Equation (Equation 31), Lemma 1 illustrates the trend of this probability with different robustness coefficient Γ.

**Lemma** **1.**
*Let θ≥0, for $\Gamma =\theta \sqrt{m}$:*

$$\lim_{m\to \infty }P\leq 1-\Phi \left(\theta \right),$$

*where*

$$\Phi \left(\theta \right)=\frac{1}{\sqrt{2\pi }}\int_{-\infty }^{\theta }exp\left(-\frac{y^{2}}{2}\right)dy$$

*is the cumulative distribution function of a standard normal. The De Moivre–Laplace approximation of the binomial distribution can be used here to obtain the approximation that applies even when Γ does not scale as $\theta \sqrt{m}$, which is expressed as follows.*

(32)
$$P\leq 1-\Phi (\frac{\Gamma -1}{\sqrt{m}}).$$



The proof of Theorem 3 and Lemma 1 can be derived along similar lines to those in Bertsimas [34], Theorem 3. Equation (Equation 31) characterizes the upper limit of the violation probability when employing the proposed OR solution with the assumed Γ. From Equation (Equation 32) in Lemma 1, it can be observed that as robustness coefficient Γ increases, $\Phi (\frac{\Gamma -1}{\sqrt{m}})$ also increases, and the probability value of constraint violation gradually decreases. The larger the value of Γ, the less the chance of violating overtime constraints and vice versa. The numerical result shown in Section 4.1 also validated our analysis. With this quantitative relationship between the bounded constraint violation probability *P* and Γ, decision-makers can choose the value of robustness coefficient Γ according to the desired constraint violation probability. Moreover, the reliability of the OR scheduling solution can also be verified by calculating *P* of the next-day OR scheduling model when the robustness coefficient is Γ.

## 4. Modeling and Analysis

To verify the validity of the proposed OR robust scheduling model and the influence of the robustness coefficient Γ on the model’s results, we conduct numerical experiments on the processed robust model in this section. We also calculate the upper limit of constraint violation probability and sensitivity analysis about the perturbation factor.

We set the relevant parameters for this paper as follows. We consider a department with 20 patients waiting for surgery on the next day and five ORs are available, and the patient’s waiting sequence can include different types of patients. We consider OR working hours of 9:00–17:00 and a fixed daily working time, T=8. Similarly to Wang [22], we assume that the uncertain surgery duration $d_{i}$ for different types of patients is generated via a lognormal distribution with the same mean value and standard deviation. In contrast, the mean value obeys a uniform distribution within [1,3]. To better show the sensitivity of uncertainty on the total OR management cost, we introduce an uncertainty perturbation factor $\delta_{i}$, which is defined as $\delta_{i}={\hat{d}}_{i}/{\bar{d}}_{i}$. To facilitate the discussion of the impact of OR fixed opening cost $c_{j}$ on the total OR management cost, two sets are settled as follows: $\left\{c_{1}=8,c_{2}=8,c_{3}=8,c_{4}=8\right\}$ labeled as Cs and $\left\{c_{1}=8,c_{2}=6,c_{3}=8,c_{4}=6\right\}$ labeled as Cd. Other relevant parameters are set as follows: The waiting penalty factor $b_{i}$ (*i* = 1,…,20) for patient *i* is taken randomly within [1,3], and the unit overtime cost is taken as $r_{1}=2,r_{2}=2,r_{3}=2,r_{4}=2,r_{5}=2$. All the following numerical results are calculated by conducting Monte Carlo experiments over 1000 realizations of the uncertain surgery duration $d_{i}$. We solved the standard MILP model in (Equation 30) on the Microsoft Visual C++ 2010 x64 platform using Python to call solver Gurobi (Version 9.1.2). All computations are carried out on a computer equipped with a dual-core 3.40 GHz Intel Core i7 processor and 16 GB of RAM. For the parameter values set in the previous paragraph, the OR scheduling problem takes approximately 160 s per run.

### 4.1. The Performance Analysis against the Robustness Coefficient Γ

Based on these computations, we perform a detailed analysis of the results.

The constraint violation probability *P* versus different robustness coefficient Γ is provided in Figure 2. From Figure 2, we can see that when robustness coefficient Γ is 0, and the constraint violation probability *P* is the largest, i.e., 59.6%. As robustness coefficient Γ increases, constraint violation probability *P* decreases. This is consistent with the description provided in Section 3.3. Robustness coefficient Γ indicates the degree of conservatism of the model considering the uncertain surgery duration. As Γ increases, the robust OR scheduling model becomes conservative, and naturally, the guaranteed constraint violation probability *P* decreases.

Figure 3 shows the curve of the minimum total OR management cost versus the robustness coefficient Γ for the different scenario with perturbation factor $\delta_{i}=0.4$. When Γ varies from 0 to 3, the minimum total OR management cost curve with Cs is the steepest, which indicates that the optimal solution of the OR scheduling model is sensitive to robustness coefficient Γ at this stage. This is because as the uncertainty set, U(Γ), of the considered uncertain surgery duration, $d_{i}$, becomes larger, and more randomness is considered for $d_{i}$. Therefore, inevitably longer waiting times for the patient and longer overtime for the OR will occur, leading to a gradual increase in a hospital’s total OR management cost. As such, we conclude that robustness coefficient Γ in [0,3] is the interval on which a manager should focus on, given that a slight change can cause a massive alteration in the minimum total OR management cost output by the model. We can also observe that the OR scheduling system begins to stabilize around a specific value at Γ=3 and beyond. Moreover, when the fixed open cost varies, the curve trend in the case of Cd is essentially the same as the fixed open cost in the case of Cs, indicating that the total OR management cost’s trend does not change with a fixed opening cost.

From Figure 2 and Figure 3, it can be observed that the total OR management cost is minimized by completely disregarding the uncertainty of surgery duration when Γ=0, at this point the optimal OR scheduling cost is 89.95. However, due to ignoring the uncertainty of surgery durations, the robustness of the solution typically cannot satisfy the constraint, and the constraint violation probability *P* reaches up to 59.6%. As robustness coefficient Γ increases to 3, the minimum total OR management cost increases, and constraint violation probability *P* decreases to 33.6%. At this point, the OR scheduling system becomes increasingly conservative, which suggests that increasing the robustness coefficient Γ can increase the protection function in a corresponding proportion and then increase the OR scheduling management cost. Our model can provide a trade-off among optimality, robustness, and violation probability based on the above results.

Figure 4 shows the differences in the total OR management cost versus robustness coefficient Γ for different perturbation factors $\delta_{i}$ in the case of Cd, which demonstrates the effect of Γ on the objective function under different circumstances. It can be easily observed that both curves exhibit essentially the same trend, and the minimum total OR management cost increases with Γ. Specifically, the minimum total OR management cost is sensitive in the [0,3] interval and stabilizes after increasing to a certain point Γ=3. The [0,3] interval represents a key area on which decision makers should focus on regardless of the circumstances. When Γ=0, the two curves have smaller gaps. However, as Γ increases, the growth rate of $\delta_{i}=0.2$ is significantly weaker than that of $\delta_{i}=0.4$. Until Γ=4, these two curves stabilize at around 93.5 and 106.5. It can be seen that the cost of $\delta_{i}=0.4$ is significantly higher than the cost curve of $\delta_{i}=0.2$. This is because the larger $\delta_{i}$ is, the larger the uncertainty set U(Γ), leading to an increase in OR operating costs to guarantee the robustness of the OR scheduling system.

Figure 5 shows the plot for the total ORs overtime versus robustness coefficient Γ. Initially, the total ORs overtime increases rapidly with the increase in robustness coefficient Γ, which manifests that total ORs overtime is sensitive to Γ in interval [0,3]. The total ORs overtime is sensitive to Γ interval [0,3]. The carve starts to stabilize in the Γ interval [3,20], which suggests that choosing more values of Γ in the interval [3,20] has a relatively small effect in terms of reducing the total ORs overtime. It can be interpreted that the robust OR scheduling model becomes increasingly conservative as Γ increases. Notably, the trend of the curve for total ORs overtime with Cd is essentially similar to the curve with Cs.

Furthermore, combining Figure 2 and Figure 5, the total OR overtime is minimized by completely disregarding the uncertain surgery duration when Γ=0, but the constraint violation probability *P* is 59.6%. As robustness coefficient Γ increases to 3 and the total ORs overtime increases, the value of constraint violation probability *P* decreases to 33.6%. The above situation is because as the robustness coefficient Γ increases, the end-time of surgery assigned to each patient is later in order to ensure the robustness of the OR scheduling system; therefore, the total time that the OR needs to be open for increases accordingly.

Figure 6 shows total patients’ waiting time versus robustness coefficient Γ. We can see that the total patients’ waiting time exhibits a slight increase concerning Γ when the robustness coefficient Γ is small and stabalizes. The total patients’ waiting time is at the minimum without considering uncertainty. Moreover, the total patient waiting time increases when considering more uncertainties because more uncertainties require the OR scheduling system to allocate a later surgery ending time for each patient. After increasing to Γ=3, the total patient waiting time remains essentially the same, indicating that increasing uncertainty set U(Γ) does not affect the total patient waiting time but only reduces constraint violation probability *P*. Moreover, the above figures show that the change in robustness coefficient Γ has a lesser effect on the total patients’ waiting time than other factors.

Combining Figure 3, Figure 4, Figure 5 and Figure 6, we present some practical interpretations of the proposed OR scheduling model. Actually, robustness coefficient Γ is an indicative parameter designed to provide some insightful advice to the hospital manager. Specifically, adjusting Γ is a method that the hospital manager of hospital can use to balance the management cost and the quality of service. In particular, by controlling the desired constraint violation probability *P*, the hospital manager can choose the appropriate Γ to design the desired robust OR solution; by controlling the value of Γ, the hospital manager can control both the robustness of the OR model and *P*. From Figure 3, Figure 4, Figure 5 and Figure 6, we observed that the OR scheduling system’s cost is susceptible to robustness coefficient Γ when Γ≤3, but for higher values of Γ, the sensitivity of the system cost is relatively independent of Γ. Hence, if a hospital manager wants a more minor OR management cost, he should select a lower Γ, which also results in a larger constraint violation probability *P*. Conversely, if he pursues the reliability of the OR scheduling model, he should select a higher Γ, but a high total OR management cost necessarily results. Moreover, [0,3] is the interval of focus for the hospital manager because the total OR management cost, total ORs overtime, and total patients’ waiting time significantly change in this interval, and *P* is retained at a high level. When Γ is in [3,20], increasing Γ exhibits no substantial effects on the total OR management cost, total OR overtime, and total patient waiting time, while *P* decreases. This suggests that [3,20] is a desirable interval for a hospital manager to avoid risks for more a smaller *P*. Finally, to show the analysis calculation values of the minimum total OR management cost, the total ORs overtime, the total patient waiting time, and the constraint violation probability with different robustness coefficients, we present numerical results in Table 2.

### 4.2. Sensitivity Analysis

Additionally, we perform a sensitivity analysis of the proposed approach related to the uncertain surgery duration, and the results are graphically presented in Figure 7 with Γ=0.4. In Section 4.1, we introduce uncertainty perturbation factor $\delta_{i}$ to better show the sensitivity of uncertainty on the total OR management cost. It reflects the uncertainty degree of the surgery duration, i.e., the more extensive $\delta_{i}$, the greater the range of fluctuation in surgery duration, and vice versa. In the proposed model, we first calculate the aggregate OR management cost against constraint violation probability *P* when we have full knowledge of the surgery duration of all patients and then compare the result with the cases when $\delta_{i}$ is set as 20%,40%, and 60%. From Figure 7, it can be seen that increasing $\delta_{i}$ raises the minimum total OR management cost. This is because a larger $\delta_{i}$ corresponds to a more extensive uncertainty set when performing OR scheduling planning. Hence, it is apparent that the minimum total OR management cost obtained by our proposed model increases significantly with increasing $\delta_{i}$. This agrees with our expectation that in a more extensive uncertainty set, an increasingly conservative OR scheduling system is required to maintain the induced opening hours of each OR below the allowed threshold. However, as $\delta_{i}$ increases, the minimum total OR management cost growth decreases. This behaviour of our proposed method sheds light on the impact of the uncertainty with which the minimum total OR management cost varies with the active region of overtime constraints.

Moreover, it can be observed that the minimum total OR management cost decreases with increasing constraint violation probability *P*, which is expected because *P* serves as the trade-off parameter to adjust the service level of the robust solution. Notably, when P>0.35, the constraint violation probability *P* can significantly affect the cost compared to when P<0.2. Meanwhile, the difference between the minimum total OR management cost of different curves gradually decreases until the same minimum value is achieved. This shows that as constraint violation probability *P* increases, the OR scheduling system gradually ignores the impact of $\delta_{i}$, and the minimum total OR management cost decreases at the expense of reliability. On the contrary, when constraint violation probability *P* is smaller, the system can bear the total OR management cost resulting from the pursuit of reliability. The more uncertainty is taken into account, the less the change in $\delta_{i}$ has an important impact on the system’s cost, i.e., the larger the constraint violation probability, the more sensitive the total OR management cost to the uncertainty perturbation factor $\delta_{i}$. This agrees with our expectation that in a larger uncertainty set U(Γ), the minimum total OR management cost is obtained to guarantee the robustness of the OR scheduling system. Figure 7 shows the robustness of our proposal relative to uncertain surgery durations and its ability to sustain the operation and management of hospitals in terms of minimum costs.

## 5. Conclusions

In this paper, we proposed a robust discrete optimization model to study the uncertain surgery duration OR scheduling problem. We consider minimizing the total OR management cost consisting of the fixed OR opening cost, unfixed OR overtime cost, and patients’ waiting penalty cost. Then, we employ a box uncertainly set and robustness coefficient Γ to specify the uncertain surgery duration. To make this model solvable, we then transformed this robust model into an MILP model by employing the robust discrete optimization theory and strong dual theory. In addition, we also presented the constraint violation probability associated with overtime constraints to guarantee the reliability of this model. Finally, we used Python 3.10 programming and called the Gurobi 9.1.2 solver to solve this model. Above all, we observed the following theoretical results and management decision suggestions:

(1). Via data analysis, we observed that the larger the robustness coefficient Γ, the higher the OR overtime and total management cost, the lower the constraint violation probability *P*, and the more conservative the OR system. Therefore, a hospital manager should choose the appropriate Γ value to achieve a suitable trade-off between robustness, optimality, and reliability. Conversely, by the reasonable value of the lowest constraint violation probability *P*, the hospital manager can also reverse the choice of Γ and design an OR scheduling system that satisfies certain reliability.

(2). Sensitivity analyses show that an appropriate increase in total OR management costs can reduce constraint violation probability *P* when *P* is large. Moreover, with different uncertainty perturbation factors $\delta_{i}$, the larger the perturbation factor $\delta_{i}$, the larger the total OR management cost. This encourages the hospital manager to accurately categorize surgeries within a speciality and to record each detailed surgery duration to improve prediction accuracies (and reduce the uncertainty perturbation factor $\delta_{i}$) and, thus, the OR solution’s quality.

(3). The total management cost obtained is sensitive to small values of the robustness factor. However, it hardly changes when the robustness factor increased to a specific value. For this paper, the OR scheduling system cost is susceptible to robustness factor Γ at Γ≤3, but for higher values of Γ, the sensitivity of the system’s cost is relatively independent of Γ, and hospital administrators should choose the appropriate Γ for their needs.

The proposed model is shown to reduce the impact of the uncertain surgery duration on hospitals and patients. It also provides theoretical support and new management solutions to facilitate decision making by hospital managers. An interesting direction for future research is to explore the multi-stage OR scheduling problem and the effect of the no-show behaviour of patients and physicians on OR scheduling. Another direction for future studies is to address the OR scheduling model by considering non-homogenous ORs and the doctor’s preference.

## Figures and Tables

**Figure 1 ijerph-19-13685-f001:**
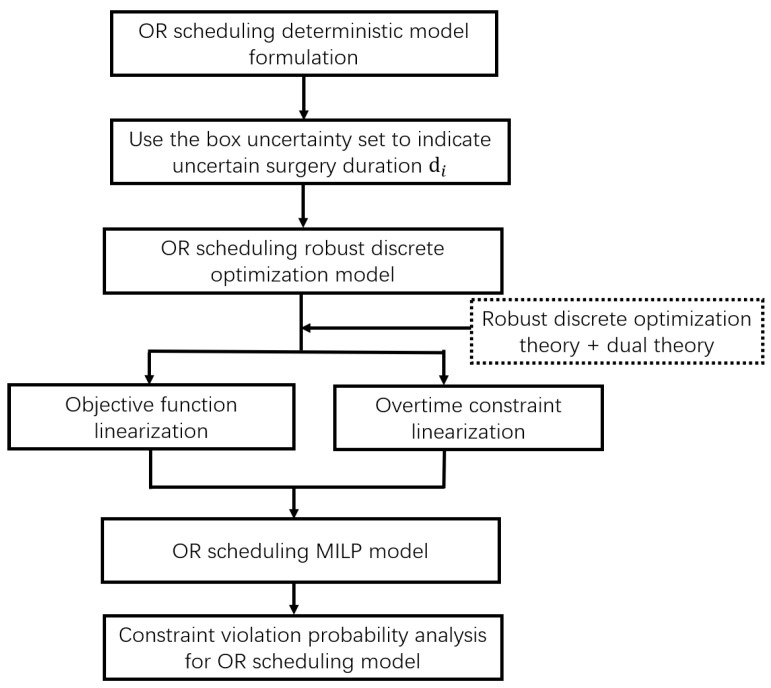
Flowchart of OR scheduling research approach.

**Figure 2 ijerph-19-13685-f002:**
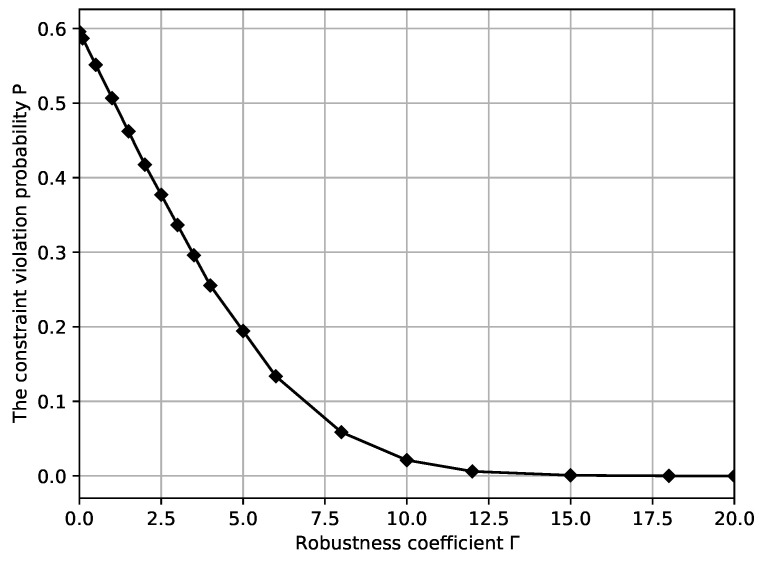
The constraint violation probability *P* versus different robustness coefficients, Γ.

**Figure 3 ijerph-19-13685-f003:**
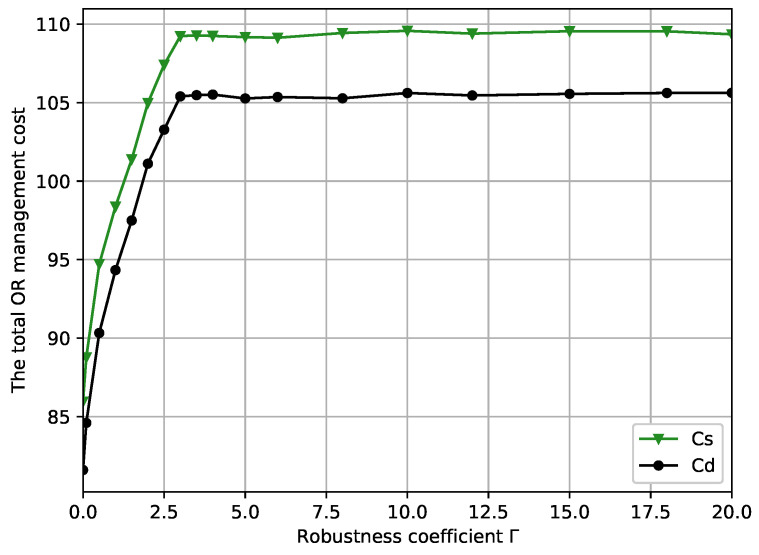
The minimum total OR management cost versus robustness coefficients, Γ.

**Figure 4 ijerph-19-13685-f004:**
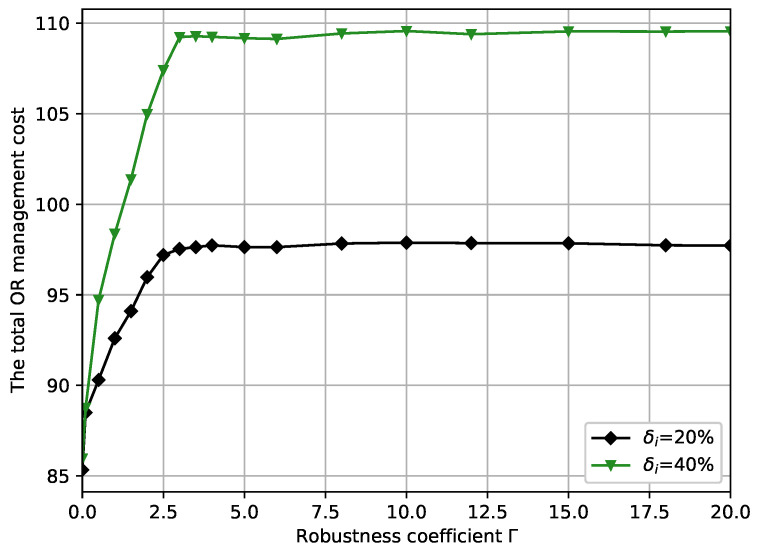
The minimum total OR management cost versus robustness coefficient Γ with different perturbation factors $\delta_{i}$.

**Figure 5 ijerph-19-13685-f005:**
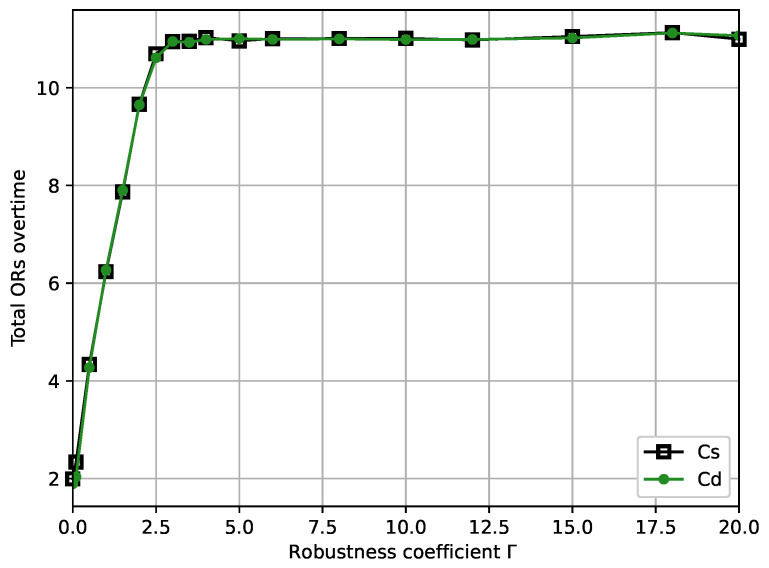
Total ORs overtime versus robustness coefficient Γ.

**Figure 6 ijerph-19-13685-f006:**
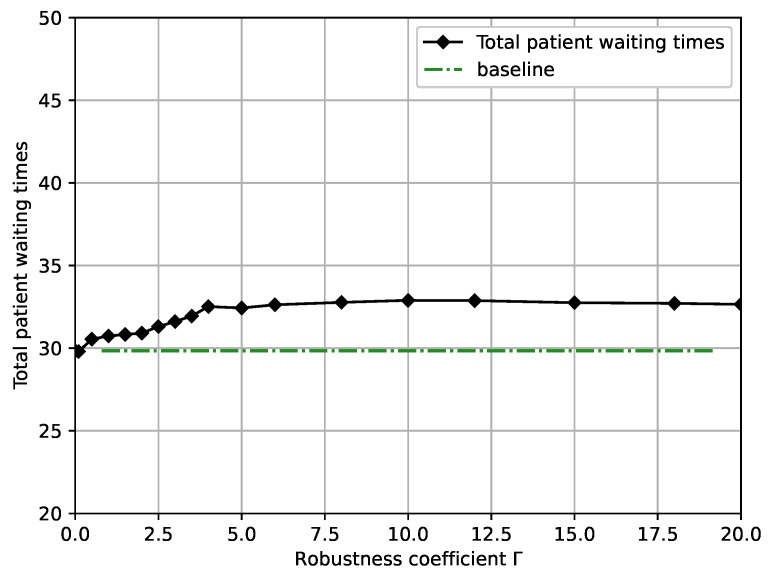
Total patients’ waiting time versus robustness coefficient Γ.

**Figure 7 ijerph-19-13685-f007:**
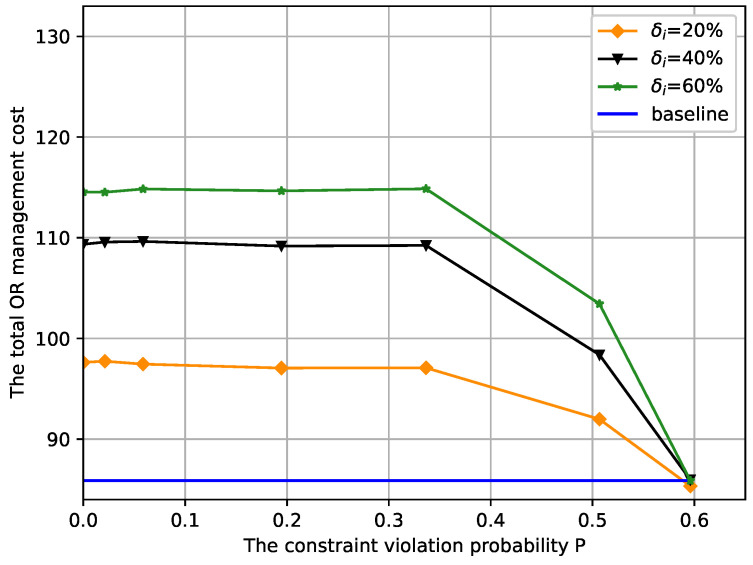
The total OR management cost versus constraint violation probability *P* under different perturbation factors $\delta_{i}$.

**Table 1 ijerph-19-13685-t001:** Symbols and notations used in this paper.

Parameters	Descriptions
M	the set of patients who need to undergo surgery on the next-day.
$d_{i}$	the expected duration of patient *i*’s surgery.
$b_{i}$	*i*-th patient’s waiting time weight, we carve out per patient’s waiting cost by the product of $b_{i}$ and per patient’s actual waiting time.
N	the OR set, in which OR is functionally identical.
$c_{j}$	fixed opening cost of the *j*-th OR.
*T*	fixed daily opening hours for each OR.
$r_{j}$	unit overtime cost.
*P*	constraint violation probability.
$x_{ij}$	decision variable, 0−1 variable, if *i*-th patient is operated in the *j*-th OR, $x_{ij}$ = 1; otherwise, $x_{ij}=0$.
$y_{j}$	decision variable, 0−1 variable, if *j*-th OR is open on next-day, $y_{j}=1$. otherwise $y_{j}=0$.
$o_{j}$	decision variable, the overtime of *j*-th OR, $o_{j}\geq 0$.

**Table 2 ijerph-19-13685-t002:** Minimum total OR management cost, total OR overtime, total patient waiting time, and the constraint violation probability under different robustness coefficient Γ.

Robust Coefficient Γ	Minimum Total OR Management Cost	Total OR Overtime	Total Patients’ Waiting Time	Constraint Violation Probability
Γ = 0	85.95	1.99	29.84	0.59
Γ = 0.1	88.76	2.33	29.79	0.58
Γ = 0.5	94.71	4.33	30.69	0.55
Γ = 1	98.37	6.23	30.63	0.50
Γ = 1.5	101.37	7.86	30.52	0.46
Γ = 2	104.97	9.66	30.50	0.41
Γ = 2.5	107.40	10.69	31.20	0.37
Γ = 3	109.23	10.94	32.64	0.33
Γ = 4	109.24	11.00	32.61	0.25
Γ = 5	109.17	11.02	32.43	0.195
Γ = 6	109.13	11.03	32.52	0.13
Γ=8	109.63	11.06	32.77	0.05
Γ=10	109.87	11.09	32.89	0.02
Γ = 15	109.54	11.11	32.75	0.0008
Γ = 18	109.74	11.12	32.71	0.00002
Γ = 20	109.35	11.13	32.65	0 ^1^

^1^ The data in this table were obtained at perturbation factor $\delta_{i}$ = 0.4 with $C_{s}$.

## Data Availability

Not applicable.

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
