# Peer review of "Robust Operating Room Scheduling Model with Violation Probability Consideration under Uncertain Surgery Duration"

_ijerph, 2022, doi:10.3390/ijerph192013685_

Round 1

Reviewer 1 Report

  1. Abstract needs to be re-written and please also address following ones: (i) State the key results/findings of proposed work. (ii) State the percentage improvement by proposed work, compared with existing works.
  2. The introduction of the paper requires significant re-writing and re-organization to really point out the motivation. If the problem is not well established, it will be very difficult or even impossible to show the contribution of the study. The introduction should be re-write and re-organize as follows, each of the stages in a separate paragraph: i. The background information (paragraph 1). ii. A general problem (paragraph 2) iii. Why the proposed approach is so important? (paragraph 3) iv. Point out limitations of the previous approaches. (paragraph 4) v. Provide alternative approaches with advantages over the previous ones used in the literature – justification (paragraph 5). vi. The objective of the study (paragraph 6) vii. The last paragraph should outline the paper structure (paragraph 7). viii. Summarized the contributions of the paper in bullets points (paragraph 8 - optional). With such a structure in the introduction, the contributions of the paper will be clear when the problem pointed out is solved.

  1. There are some typos and grammatical errors within the manuscript that need to be addressed.
  2. Please give a flowchart for summarizing the work.

  3. Please cite following papers:

-Surgical scheduling by Fuzzy model considering inpatient beds shortage under uncertain surgery durations

-A two-stage robust optimization approach for the master surgical schedule problem under uncertainty considering downstream resources

-Medical Data Analysis for Different Data Types

-Distributed Messaging and Light Streaming System for Combating Pandemics: A Case Study on Spatial Analysis of COVID-19 Geo-tagged Twitter Dataset

  1. The tradeoff between conservatism and robustness should be explored.

  2. The worst-case criterion should be applied when the availability of ICU beds is uncertain.

Reviewer 2 Report

The paper studies a robust optimization model for efficient OR scheduling problem. The paper is very well written and the structure is very clear and easy to follow. I have following questions/ comments, addressing which will clarify the paper and add to the merits of the paper:

1-Please elaborate what the second term mean in equation number 1.

2- I feel changing the "opening hour" phrase to working or available time is more meaningful. Opening hour is not  a good description for T.

3- Rewrite line 550. It is hard to follow. 

4- Another area of future study is to consider non-homogenous ORs and consider the doctor preference constraints. You can add it to the end of your study.

5- In line 277, why rounding down?

6- Can you please provide a more practical interpretation for Figures 2-5?

7- Please provide some information regarding the computational time as  well.

Round 2

Reviewer 1 Report

My all comments/suggestions have been addressed by the authors.